# COVID-19 health information needs of older adults from ethnic minority groups in the UK: a qualitative study

Priyamvada Paudyal ![ORCID],[1] Emily Skinner,[1] Saliha Majeed-Hajaj,[2] Laura J Hughes,[3] Naresh Khapangi Magar,[4] Debbie Isobel Keeling,[5] Jo Armes,[6] Kavian Kulasabanathan,[7] Elizabeth Ford ![ORCID],[1] Rebecca Sharp,[8] Jackie A Cassell ![ORCID] [1]

For numbered affiliations see end of article.

**Correspondence to**
Dr Priyamvada Paudyal;
p.paudyal@bsms.ac.uk

## ABSTRACT

**Objective** This study aimed to identify the COVID-19 health information needs of older adults from ethnic minority groups in the UK.

**Study design** A qualitative study using semistructured interviews.

**Setting and participants** Indian and Nepalese older adults (≥65 years), their families (≥18 years) and healthcare professionals (HCPs) (≥18 years) engaging with these communities. Participants were recruited between July and December 2020 from Kent, Surrey and Sussex through community organisations.

**Results** 24 participants took part in the study; 13 older adults, 7 family members and 4 HCPs. Thirteen participants were female, and the majority (n=17) spoke a language other than English at home. Older participants mostly lived in multigenerational households, and family and community were key for providing support and communicating about healthcare needs. Participants' knowledge of COVID-19 varied widely; some spoke confidently about the subject, while others had limited information. Language and illiteracy were key barriers to accessing health information. Participants highlighted the need for information in multiple formats and languages, and discussed the importance of culturally appropriate avenues, such as community centres and religious sites, for information dissemination.

**Conclusion** This study, undertaken during the COVID-19 pandemic, provides insight into how health information can be optimised for ethnic minority older adults in terms of content, format and cultural relevance. The study highlights that health information interventions should recognise the intersection between multigenerational living, family structure, and the health and well-being of older adults, and should promote intergenerational discussion.

## STRENGTHS AND LIMITATIONS OF THIS STUDY

⇒ This is the first study focusing on the COVID-19 health information needs of ethnic minority older adults from Indian and Nepalese backgrounds.

⇒ Participants had the option of being interviewed in their native language and dialect.

⇒ No information related to perception of COVID-19 vaccination was collected as the study was conducted at an early stage of the pandemic as part of a rapid response project on COVID-19.

## BACKGROUND

Since December 2019, there have been 252 million cases of COVID-19 confirmed worldwide, with 5.08 million deaths attributed to the disease (as of 12 November 2021).[1] The first COVID-19 cases in the UK were confirmed in January 2020, and since then the UK has had some of the highest COVID-19 mortality and morbidity rates in the world.[2] Among those most affected in the UK are individuals from ethnic minority groups. According to the Public Health England Report 2020, deaths from COVID-19 among people from minority ethnic groups were two to four times greater than those among the white population in England.[3] Findings from the Intensive Care National Audit and Research Centre indicate that 30% of patients critically ill with COVID-19 in intensive care throughout the pandemic have been from ethnic minority groups, with individuals from Asian backgrounds accounting for over half of these admissions.[4] These figures are significant considering that ethnic minority groups, and more specifically, those from Asian backgrounds, make up 14% and 7.5% of the population of England and Wales, respectively.[5]

Multiple independent reviews have found that socioeconomic inequalities and structural racism manifesting as poorer quality housing, overcrowding and factors related to employment which existed prior to the pandemic have contributed to this high and unequal COVID-19 death toll.[6–8] Findings indicate that individuals from ethnic minority groups face greater barriers when trying to protect themselves from COVID-19, leaving them more exposed to the virus than their

white counterparts. Overcrowded housing conditions, higher prevalence of work in public-facing keyworker roles, lower likelihood of being provided with personal protective equipment in the workplace, and a greater reliance on public transport have all contributed to this high risk of exposure. Once infected, individuals from these groups are more likely to suffer severe illness as a result of pre-existing conditions or barriers to accessing healthcare.[6–8]

Furthermore, older adults are disproportionately affected by COVID-19, with 75% of excess deaths occurring in those aged 75 and over.[3] Individuals aged 80 and over with a positive COVID-19 test are 70 times more likely to die than those under 40 with a positive COVID-19 test.[3] In particular, older adults from ethnic minority backgrounds are at a higher risk from COVID-19, partly due to increased exposure from overcrowded or multigenerational living, and partly due to chronic health conditions such as diabetes, high blood pressure and cardiovascular disease, all of which are more prevalent among ethnic minority groups than in the white population.[3 8]

Multiple recommendations have been made to reduce COVID-19 risk and decrease COVID-19 mortality for ethnic minority older adults. One recommendation is to improve the availability of culturally sensitive COVID-19 health information to enable ethnic minority older adults to make safe and informed health choices. At present, much of the COVID-19 health information available in the UK is provided in English and lacks cultural nuance. This study was undertaken as part of a larger study that aimed to address this gap by co-producing culturally appropriate COVID-19 health information. The study explored the social context of older adults from Indian and Nepalese backgrounds, their experience and understanding of COVID-19, and their access to healthcare, in order to guide and contextualise the information content and delivery. The details of the study can be found at www.bsms.ac.uk/co-rem.

## METHODS

A qualitative study was conducted using semi-structured interviews with Indian and Nepalese older adults (aged 65 and over), their relatives (aged 18 and over) and healthcare professionals (HCPs) (aged 18 and over) who engaged with these communities during the COVID-19 pandemic. Participants were recruited between July and December 2020 from Kent, Surrey and Sussex (KSS) through community organisations. Researchers networked with places of worship and local authorities to identify potential participants. Eligible members of the Indian or Nepalese community were identified by community leaders.

Once potential participants were identified, telephone contact was made by researchers (SM-H, NKM and ES) to raise awareness about the project, answer queries and arrange a suitable time for interview. Prior to interview, participants were provided with an information sheet

---

**Box 1    Topic guide for interview participants**

**Topic guide for older adults**
⇒ Participants' experience during the COVID-19 pandemic.
⇒ Sources and understanding of COVID-19 health information.
⇒ Beliefs and risk perception about COVID-19.
⇒ Use of preventive measures and health services.

**Topic guide for family members**
⇒ Older relative's knowledge and management of COVID-19.
⇒ Older relative's sources and understanding of COVID-19 information.
⇒ Protective and preventive measures taken by their relatives.
⇒ Older relative's experience and satisfaction with healthcare services.

**Topic guide for healthcare professionals**
⇒ Experience of working with older adults from ethnic minority groups.
⇒ Level and kind of engagement with healthcare regarding COVID-19 within the communities.
⇒ Routine challenges, barriers or benefits when working with these communities.
⇒ Support needs of the communities and way to improve communicating health messages.

---

about the research study including details regarding the purpose of the study, why they had been invited to participate, confidentiality and the voluntary nature of their participation. The participant information sheet was available in Hindi, Nepalese and English, and participants could choose according to their language of preference. Immediately prior to interview, the contents of the participant information sheet were discussed again and participants were given the opportunity to ask questions or raise concerns. Verbal informed consent was then obtained and recorded using a digital voice recorder, along with the rest of the interview.

All interviews were conducted by telephone (except one by online videoconferencing) due to restrictions designed to reduce spread of COVID-19. Key topics discussed included older community members' experience of lockdown, their knowledge and sources of information on COVID-19, their use of preventative measures and health services, and their opinions on what further health information they required. Family members' perceptions of the older individual's knowledge, understanding and feelings on the same topics were also collected. Information was gathered on HCPs experiences of working with older Indian or Nepalese individuals, the community's engagement with healthcare services during the pandemic, and the support needs of these communities (box 1). Demographic data was collected from all participants.

Participants had the option of being interviewed in Nepali, Hindi or English. Interviews conducted in Nepali (by NKM) were translated professionally into English and transcribed verbatim. Interviews conducted in Hindi (by SM-H) were translated and transcribed by the researcher, and interviews conducted in English (by SM-H and ES) were professionally transcribed. Indian and Nepalese interview scripts were checked for accuracy by the research team. Once transcribed, thematic content analysis was undertaken to analyse the data in NVivo (V.10)

**Table 1**  Participant demographic information

| | |
|---|---|
| **Gender** | |
| Female | 13 |
| Male | 11 |
| **Age** | |
| 75–84 | 2 |
| 65–74 | 11 |
| 55–64 | 3 |
| 45–54 | 3 |
| 35–44 | 4 |
| 25–34 | 1 |
| **Education** | |
| No school | 2 |
| School leavers | 5 |
| College leavers | 3 |
| A-Level | 3 |
| University | 11 |
| **Language spoken at home*** | |
| Gujarati | 5 |
| Punjabi | 1 |
| Hindi | 2 |
| Nepalese | 9 |
| English | 3 |
| **Number of generations living in the same household*** | |
| Three | 10 |
| Two | 4 |
| One | 6 |
| **Religion*** | |
| Jain | 2 |
| Hindu | 9 |
| Sikh | 3 |
| Buddhist | 3 |
| Not disclosed | 3 |
| **County*** | |
| Kent | 8 |
| Surrey | 7 |
| Sussex | 5 |
| **Ethnic background** | |
| Nepalese | 12 |
| Indian | 12 |

*Information not available on HCP (n=4).
HCP, healthcare professional.

using Burnard's method.[9] Burnard's method is designed for use with qualitative data collected by semistructured interviews which have been audiorecorded and transcribed. Data are systematically categorised and coded, highlighting and linking themes within the texts.

Consideration was given to the ability of the researchers involved in interviewing participants (SM-H, ES and NKM) to manage and mitigate unforeseen discomfort or distress during the interviews. The researchers attended Good Clinical Practice training prior to the interviews. ES, as a paramedic practitioner, and previously as a care worker, has worked extensively with other HCPs and with vulnerable individuals. SM-H has extensive experience conducting qualitative research with vulnerable populations, especially with refugees and asylum seekers. NKM has a background in journalism with years of experience conducting interviews with older adults, especially from a Nepalese background. To ensure reflexivity and rigour, field notes were kept, and concerns were discussed during team meetings.

### Patient and public involvement

Two community members from Nepalese background were involved in the overall study design from the start. The members contributed to the refinement of the interview guide (advised on comprehensiveness of the interview guide and clarity of the language) and discussion on the relevance and importance of the key findings from the study.

## RESULTS

Twenty-four participants took part in the study; 13 were older adults from ethnic minority communities (6 Nepalese and 7 Indian backgrounds), 7 family members (4 Nepalese and 3 Indian) and four HCPs (2 Nepalese and 2 Indian). More than half were female (n=13), 11 had attended university and most (n=17) spoke a language other than English at home (table 1). Of the four HCPs interviewed, there were two general practitioners (GPs), one surgeon and one nurse.

All except two interviews with Nepalese participants were conducted in Nepalese, two interviews with Indian participants were conducted in Hindi and the rest were conducted in English.

### Emerging themes of health information needs

Four themes and 10 subthemes were identified in the data analysis. These were: (1) COVID-19 awareness, understanding and health information needs, (2) The living, social and community context in which people receive, discuss and action their health information, (3) Interaction with healthcare, and barriers and facilitators for receiving health information and (4) Impact of COVID-19 on health information needs and self-care strategies (figure 1).

### COVID-19 awareness, understanding and health information needs

Participants' knowledge of COVID-19 varied widely. Some spoke confidently and at length about the subject, while others had limited knowledge to share. Most knew about some aspects of COVID-19 but stated that they would

**Figure 1** Themes and subthemes identified from the data analysis.

benefit from more awareness-raising through information, or clarification, about other areas.

> Uncle is totally aware [of COVID-19 prevention, identification and management], and aunty knows a little bit, but uncle totally guides to her. Thus, they are totally aware about it (Male, Family member)

> I don't know. They say so many people have died from Corona? I don't understand. Don't know (Female, Older adult)

Participants spoke about their perception of risk to themselves of getting COVID-19 as older or ethnic minority individuals, and their experiences of friends or family members getting COVID-19.

> I feel it [the SARS-CoV-2 virus] does not bias anyone…yeah…the elderly and the ones with low immunity might get infected faster…as the elders might get [COVID-19] due to low resistance to the virus…but I don't think it will spare anyone (Male, Older adult)

Risk perception was frequently influenced by the older adult's sense of identity. One of the HCPs noted that, early in the pandemic, they had found that members of the Nepalese community believed they would be stronger than the virus, purely due to being from Nepalese or Gurkha backgrounds.

> …especially elder couples, they were saying, 'oh for Nepalese people, we are strong, we are Gurkha, there is nothing much going to happen in terms of virus wise'… I think it did take them a little bit of time for them to understand what is the consequences of the virus (Female, HCP)

Participants reported receiving information on COVID-19 from a wide variety of sources. These included television, radio, newspapers, email, WhatsApp, Facebook, Viber and YouTube. Some had actively researched online, reviewing government websites and academic journals, to inform themselves so that they could better support others. Nepalese participants reported listening to the British Forces Broadcasting Service, while Indian participants listened to Indian radio stations in order to hear news in their own language.

> I do listen to the radio, Indian, there is one channel coming up in Indian, and they are very updated with the world news also, and they explain very nicely (Male, Older adult)

Many participants raised concerns about the reliability of some information. Family members and some older adults shared doubts about their ability to understand which information was accurate and which was not.

> On WhatsApp there are lots of different fake news and fiction and he [Father] would say—'Look at what this doctor is saying and look what is going on in Italy'. I would say—'Some of the videos are not real at all'; they were sort of old videos or something and I think that put the fear into him even more. I just said to Dad—'We need to make sure we look on the website, read the information online and we look on the BBC or a valid source (Female, Family member)

> "Some people send videos, and you get one message, others send a different video and then you get different messages from that. For those of us sat at home, how do we know what is true and what is false?". (Female, Older adult)

Participants stated that they would prefer to approach their doctor to ensure the reliability of the information.

> I would take the advice of the professional of that area. I will not take any independent advice from who are in not that same facility (Male, Older adult)

Four of the 10 Nepalese participants stated that they rely on information from family and friends in Nepal. One Indian participant mentioned that he actively encourages his mother to follow advice on the UK government website 'rather than looking on social media or WhatsApp messages from some Swami in India'. One HCP explained that older Indian and Nepalese adults were receiving information from their Mother countries that was unlikely to be representative of the COVID-19 situation in the UK. This could result in their underestimation of risk or unnecessary anxiety depending on the situation abroad.

> …they listen to all these Indian channels but a lot of it is very much focused on what is going on in India. So, people feel that … I suspect the difficulty is some people might feel that whatever is going on in India reflects what goes on in England. But I don't think that is a true representation because obviously the Indian community is very different, the population is very different; their standards of health and living is very different compared to the UK (Female, HCP)

The availability of language-appropriate information was important to individuals from both backgrounds, although more so to Nepalese individuals who reported that information in Nepalese is not as widely available as it is in many Indian languages.

> It would have been a lot easier if it [COVID-19 health information] was available in Nepali. As we are also not perfect [in English], even me. So, in that case, it would have been easier for us as well as for father and mother (Female, Family member)

Participants suggested a multifaceted approach for distributing COVID-19 health information. Culturally appropriate methods of distribution, such as events at community centres and places of worship, were discussed as important avenues.

> I think the best way [to distribute COVID-19 information], one of the main ways forward would be just have a website. It can be surely in multi-languages, that's absolutely brilliant (Male, Family member)

> If something [health outreach] was organised at the temple, they [Indian older adults] will definitely come because it is a known surrounding for them (Male, Older adult)

> …they [community members] belong to a certain community group…and I suppose that if we contact the Chairman or the President of the, of these groups and disseminate the information through them (Male, HCP)

### The living, social and community context in which people receive, discuss and action their health information

Multigenerational living was common among the older adults and family. Many described mutually supportive environments with strong intergenerational relationships, where members of each generation benefitted from practical and emotional support.

> I've got very good family, my daughter, we're always together wherever we can. But if the circumstances for working, doing a job, you can't see each other, at least still we smile at each other, we try to be [supportive], find out how the day has gone (Male, Older adult)

Older adults not living in multigenerational households commonly had younger family members nearby who provided varying levels of support.

> My immediate family—they've done the shopping and the necessity of looking after [us] as well just to make sure we're okay. Keeping the communication going [with us] all the time (Female, Older adult)

While younger generations provided support to older adults with shopping, technology, understanding health information and navigating the healthcare system, older adults supported younger family members with childcare. Older adults expressed a high level of joy at spending time with grandchildren, and those not living in the same household as younger family members conveyed great sadness at being separated from them during lockdown.

> The only thing is I miss my family, that's when I get down, I wanted to see my granddaughter and I couldn't, I wanted to hold her and I couldn't. That's what, yes that's what I missed (Female, Older adult)

In addition to the family, multiple older participants reported receiving offers of support from community members or members of their faith groups at the beginning of the pandemic. They appreciated these offers and were glad there was support available for isolated individuals without family.

> The young generation in the [community organisation] really helped people, if people needed anything, they would call the youth at the [community organisation] and say they needed XYZ…they [younger people] made it clear that they wanted to help and they shared their number (Female, Older adult)

Many older adults reported that their first point of contact when reaching out for support was within the family or the wider community.

> I think most people reach to those closest to them in that way before they seek any assistance from the authorities or the social services (Male, Older adult)

Participants reported varying levels of involvement in community organisations, and some identified strongly with these roles and talked at length on the subject. Several participants described their faith group as an integral part of their life. Many were frequent attendees at their place of worship prior to lockdown and several had taken the opportunity during lockdown to explore their spirituality more extensively, using online resources to access religious or spiritual services and meet ups.

> The faith group they have all the Zoom sessions … you're able to see people rather than just talking to them over the phone. You can see them as well and you can smile and share your feelings (Female, Older adult)

### Interaction with healthcare, and barriers and facilitators for receiving health information

Participants reported varied experiences when accessing healthcare before and during the COVID-19 pandemic. Some participants reported receiving a high level of care from the National Health Service (NHS).

> My last contact was the telephone call with the Consultant and I can't praise him enough. I tell you they're a God to me and they would do anything for me, and when they do talk to you, they do talk to you as a person and not just a patient, and they take

interest in you quite a bit, how is your life going on and all that (Female, Older adult)

Others were less satisfied with the care

One thing which I am sad is, there isn't good check-up here. Even when you say that you have such and such illness, they don't care much (Male, Older adult)

A few family members and HCPs felt that, in some instances, dissatisfaction with healthcare services might be due to high expectations rather than poor care.

She's on that BAME mentality of previous generations, where because you're the eldest you get the best service (Male, Family member)

HCPs broadly felt that services being provided to ethnic minority older adults were satisfactory, although one talked extensively about systemic racism, raising concerns about unidentified barriers to accessing healthcare and a lack of targeted outreach or services addressing the needs of these communities.

Even knowing that they have access to healthcare, I think most people do know that but there are probably some people who don't quite know how to navigate the system, people who fear interrogation, passport checks, questioning, deportation even if that is not relevant. There are lots of cases of people who are eligible for care but don't seek care because they fear somehow the system is going to undo them (Male, HCP)

Older adults reported difficulties accessing healthcare services. Language was a key barrier, with poor access to reliable interpreting services reported by older adults and HCPs.

If you don't have your family with you, it becomes difficult for these elderly people who have got limited communication skills (Male, HCP)

Beyond the need for better access to interpreting services, participants also identified a need for easier access to urgent and non-urgent GP appointments, regular home-testing for COVID-19, proactive welfare checks from primary care HCPs and Council volunteers trained to give advice about COVID-19 guidelines, information on how to navigate the healthcare system, and better continuity of care between services.

The main thing is the phone [at the GP surgery] does not get picked up…we do get appointments…but we get very late appointments (Female, Family member)

They [Council volunteers trained to give advice about COVID-19 guidelines] should be calling in with various languages, so that when they call you and you feel comfortable that there's somebody there who talks your language, and you feel at ease (Female, Older adult)

Extensive discussion among older adults and family was around the need for *cultural understanding* from HCPs. Opinion varied on whether it was the responsibility of the HCP to understand the culture of ethnic minority individuals or not.

She's [Mother] an Orthodox Sikh, so she's, there's places where she won't take her dupatta off, and where, I've been with her in the past, and they've asked her to remove it, and she hasn't wanted to, and they don't understand why (Female, Family member)

I don't feel the onus lies just with the medical profession. The onus lies with the [Indian] individuals as well to try and understand that they have got to change their ways as well (Male, Older adult)

## Impact of COVID-19 on health information needs and self-care strategies

Participants reported various challenges at the beginning of the pandemic and explained how they had adapted to lockdown life. Some described a sense of uncertainty and disruption.

It was challenging because it was thrust upon us and we were not prepared for it…there was a lot of change work wise and a lot of the personal life had to adapt (Male, Older adult)

A few participants talked about isolation. The onset of shielding meant some had been unable to have domestic support and were unable to do tasks such as cleaning or putting away groceries which had been delivered to the front door.

I can't do my household chores or the carers were coming in…it was really difficult for me… well for three months, nobody came in, not even with the mask. So that was a very, very difficult time (Female, Older adult)

Although this experience was limited to a small number of participants, many commented on how it could have been if they had not had family support available and voiced concerns about those left without a support network.

I think if you've got family members and you've got their support, that's fine, but I think all people haven't got that, or are living with family but are still, haven't got that support network, I think they're the ones that need those [COVID-19 health] messages (Female, Family member)

HCPs reported challenges with inability to provide the normal level of care to their patients and fear about bringing the virus home to their families.

I just like seeing people and it [telemedicine] is not the same; I am just not practicing good medicine at all. I can't because, I don't think you can (Male, HCP)

For many participants, the pandemic and lockdown caused much anxiety. Participants were anxious about many things including catching COVID-19, wearing masks, seeking or being required to take a test, their existing comorbidities, others not social-distancing adequately, being separated from loved-ones and going out.

> We have become old….and when they [the Government] repeatedly say not to go out…we feel scared… (Female, Older adult)

Many participants talked about self-care activities, some through exercise like walking and yoga, and some through household tasks or spending time playing with grandchildren.

> For the first three months, I just did exercises at home, my wife and I. She did quite a lot of yoga and I have a gym at home (Male, Older adult)

> My daughter keeps him [Father] quite active. She keeps him on his toes quite a lot! She is like—'Grandad take me here'. In our garden he is constantly with her or constantly in our little porch area. She must keep him quite active (Female, Family member)

Multiple participants talked about maintaining good mental health and keeping up morale.

> …everything is in your mindset, how you set up your mind. Because if you think that 'I don't want to talk to this person', 'I don't want to listen anyone', 'I'll be just in my room all the time', which is not good for yourself, not good for your family, not good for anyone actually, because if you don't talk you won't come close to each other (Male, Older adult)

Participants talked about eating healthily, staying hydrated and using traditional remedies. While most participants indicated these practices were part of a multifaceted approach to staying well, a few felt that traditional remedies would be useful in the first instance to cure or prevent COVID-19.

> I don't think it [ginger and turmeric] cures, but it might help to some extent…If you follow it, it isn't bad for your health…However, we can't confirm it with a full confidence (Male, Older adult)

> Firstly, we would try our own Asian remedies for coughs and fever as this is what we tend to do when we are sick. I wouldn't go to a hospital. Our Asian remedies have a better effect than medicines (Female, Older adult)

Others reflected on the opportunity to explore their spirituality and spend more time with loved ones.

> It has, it's been very challenging in a way. Being a spiritual person, I had the opportunity to sit down and find out more about my faith, that is what I found very interesting and actually growing as well (Female, Older adult).

## DISCUSSION

To our knowledge, this is the first study with the specific aim of providing empirical evidence to guide the development and delivery of COVID-19 health information to ethnic minority older adults from Indian and Nepalese backgrounds. Our study found that multigenerational living is common among ethnic minority households, and family and community are key for receiving support and communicating about healthcare needs. Participants exhibited widely differing levels of knowledge regarding COVID-19 and preventative measures for the disease. Language and illiteracy were cited as key barriers to accessing health information. Participants highlighted the importance of information provision in multiple languages and formats, including culturally-appropriate avenues for information dissemination. The interconnectedness of community members and importance of intergenerational relationships mean that full community engagement was seen as the most effective way to develop and deliver health information to the older generation. Targeted health information interventions should recognise this intersection between intergenerational living, family structure and health, and promote intergenerational discussion to communicate health messages.

### Health information needs

Findings from this study highlighted four core areas in which health information could be targeted for greater cultural-sensitivity and effectiveness. The study found that a range of information is required to target different knowledge levels of the participants and that the message should be clear and simple in both style and content to ensure that those with limited COVID-19 knowledge have access to the required information. Targeted and culturally appropriate COVID-19 health information for ethnic minority groups has been endorsed by multiple reports and studies.[8 10–13]

Difficulty in assessing the reliability of COVID-19 information was a concern raised by many participants. Misinformation was prevalent during the pandemic, especially on social media platforms.[10 14] Targeted information is required to address the myths and misconceptions that are perpetuated, and reduce their spread. Education should be provided to the communities around how to identify reliable sources of information. This could include the recognition of logos used by official sources, such as the NHS, the government and Public Health England, with explanation regarding the trustworthiness of these sources. Further work could be done to help community members identify trustworthy sources of information within their communities, and to explore what factors make these sources reliable.

Our study shows that ethnic minority older adults often seek information from international sources, such as radio stations, television channels and YouTube videos from their native country. Although it is beneficial for them to receive COVID-19 information in their native language,[10 12 15 16] awareness needs to be raised about the

context of this information. Older adults may not realise that information gained from international sources is not always relevant to their situation in the UK, as the political, social and healthcare systems are likely to differ from the systems in their native country.

A further area vulnerable to misinformation is the use of traditional preventative measures and remedies.[10] Many participants in this study mentioned these measures which included drinking warm or boiled water infused with combinations of ginger, honey, lemon, black pepper and turmeric. Although most understood the usefulness and limitations of these measures, some reported that they would rather use them than seek medical support, and a couple of participants reported that their faith would protect them from COVID-19. These findings indicate a need for inclusion of this subject in COVID-19 health information that targets these communities.

As in the previous studies,[10 15 17] multigenerational living was common among participants in this study. The risk of COVID-19 household transmission for those living in multigenerational households intersects with a lack of financial support for isolation and inability to work from home.[18] During the pandemic, multigenerational living has been considered a challenge for ethnic minority households due to difficulty with social distancing and increased exposure to COVID-19, and has been portrayed negatively in the literature[10 13 15] and by mainstream media.[17] Only one previous study[17] has highlighted the benefits of multigenerational living and discussed ways in which families worked together in order to mitigate risk. The study found that as long as preventive measures were adopted, living in close proximity was beneficial as younger family members were able to provide care and solidarity to the older adults. Participants in the study reported mutually supportive environments with strong intergenerational relationships, but also difficulties social distancing, and increased exposure to the virus from working-age and school-age household members. While COVID-19 health information needs to highlight the risks associated with multigenerational living, it should also acknowledge the benefits of living in extended family groups and provide practical advice regarding protective behaviours to mitigate risk. This will hopefully result in information that is more relevant to ethnic minority groups and results in a higher level of engagement than advice which does not address these issues.

Findings from this study supported the development of targeted COVID-19 health information rather than generic information that aims to address the needs of all ethnic minority groups. Hanson *et al*[17] highlighted friction within and between ethnic minority groups, arguing that they should not be treated as one homogeneous group. Conversely, Ala *et al*[11] advised the development of COVID-19 information that was broadly suited to the needs of all ethnic minority groups. However, most studies recommend that health information should be targeted at individual groups.[8 10 12 13]

One HCP from this study noted that participants' perceptions of COVID-19 risk were linked with identity. Although this was noted early in the pandemic, and the acquisition of further knowledge and experience is likely to have resulted in a change of attitude, it is noteworthy that this strength of identity and cultural pride among ethnic minority groups has the potential to be either to the benefit or detriment of public health messaging.

Participants talked extensively about the activities which they undertook during lockdown. Many participants were positive about this time period but some reported anxiety, and deterioration of their health. COVID-19 health information targeted at ethnic minority older adults should not only contain information relating to COVID-19, but should also consider the mental and physical health impact of the entire pandemic and its containment measures. Smith *et al*[16] advised the development of psychoeducation packages for ethnic minority groups to ensure that mental health beliefs and knowledge are based on evidence, and address culturally grounded explanatory models.

### Development and delivery of health education resources

Similar to our study, several others have highlighted the importance of engaging with community and faith leaders in the development and delivery of health education resources.[10 11 13–15 17] One study found that many of their participants had turned to religious scriptures for guidance in the early days of the pandemic, while participants in another study expressed uncertainty about whether or not the COVID-19 vaccine was in line with their religious teachings.[15] These are issues about which trusted leaders can provide clarification.

Some participants reported that they preferred to receive health information from HCPs rather than from friends or family. However, the literature regarding the provision of health education by HCPs is mixed, and as this study identified, not all participants were having regular contact with HCPs during the pandemic. Some studies indicated that HCPs were considered trusted sources of information,[11 12 14] while others reported that ethnic minority groups were distrustful of HCPs as a result of misinformation spread during the pandemic, personal experiences of racism, and the legacy of historical abuses of power in healthcare and medical research.[19 20] Only one older adult in this study highlighted these types of concerns, citing racism as the reason for receiving poor care during a recent hospital visit. Therefore, it is unclear whether the impact of historical and contemporary racism on care differs between ethnic minority groups or if the exploration undertaken by this study lacked enough depth to expose these concerns. Instances where participants from other studies reported distrust of HCPs and the healthcare system tended to relate to concerns regarding the COVID-19 vaccine,[21 22] a topic which this study did not cover. Regardless, it is important that HCPs delivering COVID-19 health information engage with ethnic minority groups in a culturally sensitive manner

and address their sources of anxiety.[22 23] The wider literature highlights an urgent need for further research on how racial discrimination impacts the health outcomes of ethnic minority communities, particularly in relation to housing, health services, employment and the criminal justice system.[24] Funding to strengthen access to social security, adequate and secure housing, raise and widen eligibility for statutory sick pay, and abolish no recourse to public funds have been recommended to mitigate the impact of social and economic inequalities.[24]

Participants in this study expressed a wide variety of preferences when it came to receiving health information. Therefore, the delivery of health messages should be multifaceted to ensure that the information reaches everyone. This should include television, radio, newspapers and leaflets. A common way that participants liked to receive information, and keep in contact with friends and family, was through social media platforms such as WhatsApp, Viber and Facebook. Clear and simple visual health messages could be developed in the form of memes, short videos or animations for easy sharing among family members and friendship groups. This would ensure that key messages are received by a large number of people. Such an approach is supported by a recent scoping review involving the findings from 81 studies, which suggests that social media platforms can have a crucial role in tackling infodemics, misinformation and rumours in relation to COVID-19.[25] Another review also highlights the need to embrace social media by clinicians and institutions to help combat misinformation in the time of COVID-19 pandemic,[26] a view that is in line with some of the concerns raised in this study.

Box 2 summarises the recommendations for practice based on the findings from this study. These recommendations provide broader insights into how health information can be optimised for ethnic minority older adults in terms of content, format and cultural relevance.

### Study strengths and limitations

This is the first study focusing on the COVID-19 health needs of ethnic minority older adults from Indian and Nepalese backgrounds. Another strength of this study is its exploration of the opinions of diverse participant groups, consisting of older adults, their family members and HCPs to elicit rich data on the information needs of this group. However, it also has a number of limitations. Only two ethnic groups were included in the study, recruited solely from the KSS region. Consequently, findings may not be representative of the experiences of older adults from other ethnic minority groups, or of those from Indian or Nepalese backgrounds not living in KSS, which is predominantly a rural area. Also, none of the participants were Muslim, and the majority of the Indian participants came from two states in India (Gujrat and Punjab).

The restricted geographical area also limited the pool of potential HCPs. Extensive efforts were made to recruit HCPs using academic and community contacts, as well as

---

**Box 2   Recommendations for practice**

**Content and format**
⇒ Health information for ethnic minority older adults should be clear and simple in both style and content.
⇒ Information should be available in a variety of written, audio, and visual formats, with varying levels of detail to suit different preferences and reading levels.
⇒ Information should be translated into a wide variety of ethnic languages so that older adults can choose their preferred language.

**Cultural relevance**
⇒ Cultural practices, such as multigenerational living, should be acknowledged for their benefits as well as addressed regarding their challenges. Targeted health information interventions should recognise the intersection between intergenerational living, family structure and health and well-being of older adults and promote intergenerational discussion.
⇒ Education on how to assess the reliability of information should be provided based on an understanding of which organisations are seen as trustworthy or untrustworthy by different groups. There is a need on an ongoing basis to identify and fill the information gaps, address misinformation and develop targeted messages.
⇒ Health information should be co-produced with a representative sample of the community, including community and faith leaders, older adults, their family members, and associated healthcare professionals.
⇒ Healthcare professionals should take time to gain the trust of the community in order to successfully deliver health information. In order to facilitate access and convenience, the professionals need to liaise with community and faith leaders to engage with the communities at familiar settings such as community centres and religious sites.

---

multiple social media platforms. However, as the study was conducted in the midst of the pandemic, the researchers faced challenges recruiting the target number of HCPs. It was not possible to recruit HCPs through NHS organisations due to the time constraints of the study in obtaining health research authority approval. This made it difficult to contact HCPs working with Indian or Nepalese older adults. Only four of the desired 10 HCP participants were recruited. All were from an Indian or Nepalese background and were recruited through personal contacts of the research team and community organisations. Although they made valuable contributions, their insights into the needs of ethnic minority older adults often seemed based on their experiences with older family members, older patients from other ethnic minority groups and academic knowledge, rather than their work with older patients from Indian or Nepalese backgrounds. Recruitment through NHS organisations would have provided access to a wider range of HCPs, including district nurses and carers, who regularly visit older patients in their homes and provide hands-on care. Recruitment from a wider geographical area would also have increased the pool of potential participants. Remote recruitment and data collection were further limiting factors as those not adept with technology and more socially isolated were less likely to be recruited to the study. It is also possible that those

who opted to take part in the study, did so because they felt they had good knowledge of COVID-19.

Lastly, participants in this study were not questioned about their thoughts, feelings or plans regarding the COVID-19 vaccine as the study was conducted at an early stage of pandemic as a part of the rapid response project on COVID-19. However, research following the roll out of the vaccination programme has indicated higher vaccine hesitancy and lower vaccine uptake among ethnic minority groups than among the White population.[27] A longitudinal study involving 12035 participants found that COVID-19 vaccine hesitancy is highest among black (71.8%), followed by Pakistani/Bangladeshi respondents (42.3%), compared with White British or Irish respondents (15.2%).[28] Reasons for vaccine hesitancy include historical marginalisation and distrust of government and public health agencies due to ongoing discrimination, prior research conducted unethically with the ethnic minority groups and fears of being misled about vaccines.[29] In addition, concerns related to safety and lack of clear guidance on potential long-term effects of vaccine are cited as barriers to vaccine uptake.[29] Population-wide, COVID-19 vaccination is key to reducing the spread of the virus, and preventing severe illness and death from COVID-19.[22] However, COVID-19 health information from mainstream media has not been successful to reach various communities as the messaging often lacks cultural and linguistic relevance.[29] The recommendations from this study (summarised in box 2) will be valuable in effective communication of health information including COVID-19 vaccine uptake in ethnic minority communities.

## CONCLUSION

This study has provided detailed and rigorous empirical data regarding the COVID-19 health information needs of older adults from ethnic minority groups. The findings from this study have implications beyond COVID-19, to all areas of health promotion. Recommendations for the development and delivery of health education resources have been developed from the synthesis of this data with findings from previous research studies. In addition to being the only study to have investigated the COVID-19 health information needs of ethnic minority older adults from Indian and Nepalese backgrounds, it also adds much needed evidence to the pool of research regarding the experiences, needs and challenges of ethnic minority groups during the COVID-19 pandemic, as well as wider lessons about providing health information to this population. The study highlights the urgent need for an intergenerational approach embedded in the public health interventions and policies for older adults from ethnic minority groups, taking account of their sources of information in the UK and in countries of origin. Further research needs to be conducted to understand the health information needs of older adults from other ethnic minority groups and working-age adults from ethnic minority backgrounds to promote intergeneration discussion on health promotion.

**Author affiliations**
¹Department of Primary Care and Public Health, Brighton and Sussex Medical School, Brighton, UK
²Department of Geography, Queen Mary University of London, London, UK
³King's College London - Strand Campus, London, UK
⁴Centre for Nepal Studies, London, UK
⁵University of Sussex Business School, University of Sussex, Brighton, UK
⁶School of Health Sciences, University of Surrey, Guildford, UK
⁷Imperial College Healthcare NHS Trust, London, UK
⁸Kent Surrey Sussex Academic Health Science Network, Crawley, UK

**Acknowledgements** We would like to thank all the participants in this study. Without their cooperation, this study would not have been possible.

**Contributors** PP and JAC conceived the study. PP, JAC, DIK, JA, LJH, KK, NKM and RS obtained funding for the study. SM-H, NKM and ES collected the data. ES analysed the data and prepared the first draft under the supervision of PP, LJH and SM-H. PP prepared the final draft, NKM, JAC, DIK, JA, KK, EF and RS reviewed and edited the manuscript. All authors read and approved the final manuscript. PP is the guarantor of this study.

**Funding** This research was funded by the National Institute for Health Research (NIHR) Applied Research Collaboration Kent, Surrey, Sussex (Grant number: NIHR 200179).

**Disclaimer** The views expressed are those of the author(s) and not necessarily those of the NHS, the NIHR or the Department of Health and Social Care

**Competing interests** None declared.

**Patient and public involvement** Patients and/or the public were involved in the design, or conduct, or reporting, or dissemination plans of this research. Refer to the Methods section for further details.

**Patient consent for publication** Not applicable.

**Ethics approval** The study was approved by the Brighton and Sussex Medical School Research Governance and Ethics Committee (ER/BSMS3653/6). Participants gave informed consent to participate in the study before taking part.

**Provenance and peer review** Not commissioned; externally peer reviewed.

**Data availability statement** No data are available.

**ORCID iDs**
Priyamvada Paudyal http://orcid.org/0000-0002-6209-575X
Elizabeth Ford http://orcid.org/0000-0001-5613-8509
Jackie A Cassell http://orcid.org/0000-0003-0777-0385

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
