## [Reviewer comments · BMJ Open]

ARTICLE DETAILS

TITLE (PROVISIONAL)	COVID-19 health information needs of older adults from ethnic minority groups in the UK: a qualitative study
AUTHORS	Paudyal, Priyamvada; Skinner, Emily; Majeed-Hajaj, Saliha; Hughes, Laura J.; Magar, Naresh Khapangi; Keeling, Debbie Isobel; Armes, Jo; Kulasabanathan, Kavian; Ford, Elizabeth; Sharp, Rebecca; Cassell, Jackie

VERSION 1 – REVIEW

REVIEWER	Kadambari, Seilesh John Radcliffe Hospital, Department of Paediatrics
REVIEW RETURNED	05-Jan-2022

GENERAL COMMENTS	In the U.K, COVID-19 has disproportionality affected ethnic minorities. The reasons for this are likely multifactorial but poorly understood. This important study aims to address the health information needs of older adults from Indian and Nepalese backgrounds. I have some comments which will hopefully strengthen the manuscript: 1. Methods: page 5 lines 23 – 29 refers to the limitation in number of participants and should be moved to the discussion. Page 6 line 50 – page 7 line 8 does not seem relevant and can be removed.2. Methods: the patient and public involvement section needs to be expanded. How many and who were the representatives involved? How exactly did they inform the interview guide and discussion?3. Results: in table 2, it would be useful to include how many years the participants had been in the UK.4. Results: more details should be given in relation to the news sources the participants used (page 9 line 42 – 60). How many participant relied on information from extended family/friends in India and Nepal? How did the participants know which information to trust? Which HCP (i.e. their GP? District nurse? Someone else) would they see (as stated on page 10 line 12)?5. Results: the authors could offer more information on how information could be spread (under COVID-19 awareness, understanding and health needs section). So, would social media messages from the GP practice be useful? Or youtube videos?6. Discussion: the authors note that this study was conducted too early in the pandemic to evaluate attitudes to COVID vaccines. However, the authors should dedicate some time to how their findings can inform better vaccine uptake in ethnic minority groups. In particular, the authors should read and cite a recent study (Kadambari S et al Lessons about COVID-19 vaccine hesitancy among minority ethnic people in the UK Lancet Infect Dis 2021) that evaluated this issue and discuss how qualitative data in their
---

	study could be used to improve communication strategies and optimise vaccine uptake. 7. Discussion: None of the 24 participants were Muslim. This seems like an anomaly given that KSS has a relatively big Muslim population. Also, there are no south Indians included either and all Indian recruits seemed to have come from two states. This should be included in the limitations.
--	--

REVIEWER	Razieh, Cameron University of Leicester, Diabetes Research Centre
REVIEW RETURNED	24-Jan-2022

GENERAL COMMENTS	Line 41-43: reference for this statistic needed Make the protocol of the larger study which this study is part of available so others are able to review methodology, aims etc and how this may impact the current smaller study
--

VERSION 1 – AUTHOR RESPONSE

Reviewer 1	
1. Methods: page 5 lines 23 – 29 refers to the limitation in number of participants and should be moved to the discussion. Page 6 line 50 – page 7 line 8 does not seem relevant and can be removed.	The text has been moved to the discussion section as suggested (Please see page 20, line 3-5). The lines on page 6 and 7 have been deleted as advised.
2. Methods: the patient and public involvement section needs to be expanded. How many and who were the representatives involved? How exactly did they inform the interview guide and discussion?	The detail has been provided in the revised draft as suggested (please see page 7, line 2-6).
3. Results: in table 2, it would be useful to include how many years the participants had been in the UK.	We are unable to include this information as it was not collected during the study.
4. Results: more details should be given in relation to the news sources the participants used (page 9 line 42 – 60). How many participant relied on information from extended family/friends in India and Nepal? How did the participants know which information to trust? Which HCP (i.e. their GP? District nurse? Someone else) would they see (as stated on page 10 line 12)?	Four of the ten Nepalese participants stated that they rely on information from family and friends in Nepal. One Indian participant mentioned that he actively encourages his mother to follow advice on the UK government website 'rather than looking on social media or WhatsApp messages from some Swami in India'. We collected information on the news sources used but did not explore the perception around reliability of the information source. However, our results section highlights participants' concerns around the reliability of the resources. In terms of HCP, participants mentioned that they will approach their doctor, this information has been added in the revised draft. (Please see page 10, line 7-17 on the revised draft.)

5. Results: the authors could offer more information on how information could be spread (under COVID-19 awareness, understanding and health needs section). So, would social media messages from the GP practice be useful? Or youtube videos?	We do not have data in relation to this to present in the results section but we have added information about the role of social media in disseminating health information in the discussion section. (Please see page 18, line 26-31.)
6. Discussion: the authors note that this study was conducted too early in the pandemic to evaluate attitudes to COVID vaccines. However, the authors should dedicate some time to how their findings can inform better vaccine uptake in ethnic minority groups. In particular, the authors should read and cite a recent study (Kadambari S et al Lessons about COVID-19 vaccine hesitancy among minority ethnic people in the UK Lancet Infect Dis 2021) that evaluated this issue and discuss how qualitative data in their study could be used to improve communication strategies and optimise vaccine uptake.	We thank the reviewer for this comment, detailed information has been added in the revised draft as suggested, and the Kadambari reference incorporated into the text (now reference number 29). (Please see page 20, line 27-40.)
7. Discussion: None of the 24 participants were Muslim. This seems like an anomaly given that KSS has a relatively big Muslim population. Also, there are no south Indians included either and all Indian recruits seemed to have come from two states. This should be included in the limitations.	This information has been added in the limitation section of the revised draft as advised. (Please see page 19, line 37- to page 20 line2)
Reviewer 2	
Line 41-43: reference for this statistic needed	Reference has been added now.
Make the protocol of the larger study which this study is part of available so others are able to review methodology, aims etc and how this may impact the current smaller study	The details of the larger study can be found at www.bsms.ac.uk/corem. The study web link is provided in the introduction section. (Please see page 5, line 6.)